# A Comprehensive Study of Meat Quality and Flavor Characteristics of Different Sexes of Yanbian Yellow Cattle Using GC-IMS and LC-MS/MS Technologies

**DOI:** 10.3390/foods14183175

**Published:** 2025-09-12

**Authors:** Jinlong Tan, Baide Mu, Hongshu Li, Chenguang Li, Tingting Gao, Xiangji Meng, Changcheng Zhao, Chunxiang Piao, Tingyu Li, Juan Wang, Hongmei Li, Guanhao Li

**Affiliations:** 1College of Agricultural, Yanbian University, Yanji 133000, China; 2022010859@ybu.edu.cn (J.T.); mubaide@ybu.edu.cn (B.M.); m13044337020@163.com (H.L.); master0216@163.com (C.L.); 13704329945@163.com (T.G.); mengxiangji2004@163.com (X.M.); cxpiao@ybu.edu.cn (C.P.); 0000008801@ybu.edu.cn (T.L.); wangjuan@ybu.edu.cn (J.W.); 2Key Innovation Laboratory for Deep and Intensive Processing of Yanbian High Quality Beef, Ministry of Agriculture and Rural Affairs, Yanji 133002, China; 3Engineering Research Center of North-East Cold Region Beef Cattle Science & Technology Innovation, Ministry of Education, Yanji 133002, China; 4College of Life Science, Zhengzhou University, Zhengzhou 450001, China; zhaocc91@zzu.edu.cn

**Keywords:** Yanbian yellow cattle, volatile flavor, GC-IMS, LC-MS/MS

## Abstract

Differences in sex among cattle are associated with distinct physiological traits and endocrine profiles, which significantly influence meat quality and flavor characteristics. This study is the first to explore the effect of sex on meat quality, volatile flavor, and metabolites in Yanbian yellow cattle. Volatile and non-volatile metabolites in the *longissimus dorsi* muscle were comprehensively profiled using gas chromatography–ion mobility spectrometry (GC-IMS) and liquid chromatography–tandem mass spectrometry (LC-MS/MS). The results revealed that the fatty acid content (SFA, MUFA, PUFA) in the *triceps brachii* and *longissimus dorsi* muscles of cows was significantly higher than that of bulls (*p* < 0.05). In contrast, bulls showed elevated levels of total amino acids and non-essential amino acids (*p* < 0.05). Volatile flavor analysis revealed 35 discovered compounds, including alcohols, ketones, aldehydes, and esters. Notably, alcohols and aldehydic flavor compounds in the *longissimus dorsi* were substantially more prominent in cows, whereas ketonic compounds were predominantly higher in bulls. The study also identified key metabolic pathways—including lipid metabolism, protein and amino acid metabolism, and glycolysis—that are predominantly associated with meat quality. These findings provide a theoretical basis for further elucidating the molecular mechanisms regulating flavor formation in Yanbian yellow cattle.

## 1. Introduction

Beef is widely appreciated for its high-quality protein, essential nutrients, and favorable fatty acid profile, along with its rich content of vitamins and minerals [1]. Among the predominant cattle breeds in China, Yanbian yellow cattle from the northeastern region stand out [2]. Adapted to the harsh, cold conditions of northeastern China, this breed shows remarkable properties, such as resilience to the cold, strong disease immunity, and a capacity to thrive on roughage [3]. These properties contribute to its exceptional meat quality and unique flavor. With advances in science and technology, the primary role of cattle has shifted from draft purposes to meat and dairy production [4]. Consequently, investigating the effect of sex on the quality of Yanbian yellow cattle beef can offer valuable insights for farmers and slaughterhouses in formulating targeted breeding strategies, while also providing a preliminary foundation for comparing meat quality across different cuts and assessing its suitability for further processing.

Beef flavor significantly impacts consumer preference for and the economic value of beef, serving as both a vital sensory characteristic and an essential criterion for assessing beef quality [5]. The flavor profile of beef is influenced by a range of factors, including the cattle’s sex, breed, maturation, and diet. Differences in flavor are largely attributable to the physiological and genetic differences between male and female cattle, which manifest in their growth rates, carcass structures, and overall meat attributes. Males, predominantly influenced by androgens, exhibited enhanced protein accretion and reduced adipogenesis. Hormonal influences yield carcasses characterized by reduced adipose layers across the shoulder, back, and loin, contributing to a leaner physique with a greater proportion of premium cuts, such as the leg and shoulder, albeit at a decreased overall slaughter yield. Compared to bulls, cows exhibit a 4.21% lower proportion of muscle tissue, a 71.15% higher proportion of fat tissue, and an 18.52% lower proportion of bone tissue in the carcass [6]. Jamie Cafferky reported that sex significantly influenced meat quality parameters such as shear force, intramuscular fat levels, and loss during cooking across diverse breeds [7]. The impact of sex on meat flavor stems from two primary factors: firstly, the secretion of compounds such as free carboxylic acids and cholesterol by male sebaceous glands, which markedly reduces the aromatic quality of the meat [8], and secondly, the variations in fat content, fatty acid profiles, and amino acids between sexes, which contribute to distinct flavor nuances [8].

Volatile organic compounds (VOCs)—primarily aldehydes, esters, ketones, hydrocarbons, furans, and alcohols—are formed from flavor precursors through biochemical reactions, including lipid oxidation, Strecker degradation, the Maillard reaction, and thiamine degradation [9]. Flavor precursors consist mainly of water-soluble and lipid-soluble compounds. Water-soluble precursors include free amino acids, reducing sugars, peptides, thiamine, and organic acids, among others, while lipid-soluble precursors primarily comprise lipids [10]. Amino acids and reducing sugars participate in Maillard reactions to generate furans, furanones, pyridines, and other heterocyclic compounds [11]. The Strecker degradation of amino acids also contributes to this process by generating various flavor compounds [12].

In recent years, gas chromatography–ion migration spectrometry (GC-IMS) and electronic noses have been widely used in meat analysis. These technologies are renowned for their rapid, efficient, and non-invasive characteristics [13]. The electronic nose, adept at simulating human olfactory responses, effectively identifies volatile compounds, thereby addressing the inherent subjectivity and variability of human sensory assessments [14]. Furthermore, GC-IMS has gained traction by leveraging gas chromatography to segregate sample constituents, coupled with ion mobility spectrometry for compound characterization without necessitating prior sample preparation. This method is distinguished for its high sensitivity and rapid processing capabilities, making it suitable for identifying food flavors [15]. GC-IMS is frequently employed in conjunction with electronic nose technology, combining detailed molecular insights from GC-IMS with the broader sensory profiles captured by the electronic nose. This synergy enhances both instrumental analysis and sensory evaluation, providing a comprehensive understanding of food flavors [16]. Additionally, metabolomics has emerged as a significant player in discerning flavor compounds within meat products. Techniques like liquid chromatography–tandem mass spectrometry (LC-MS/MS) offer heightened sensitivity and the capacity for high-throughput sample analysis, thus enabling the comprehensive detection of a broad spectrum of metabolites in meat [17].

While the influence of sex on meat quality has been documented in several cattle breeds, such as Angus and Simmental, the specific effects on indigenous Chinese breeds like Yanbian yellow cattle remain largely unexplored. Most previous studies have focused on growth performance or carcass yield, with limited attention to a comprehensive assessment of meat quality, flavor chemistry, and underlying metabolic mechanisms. Therefore, this study aims to bridge this knowledge gap by providing the first comprehensive analysis of how sex shapes the meat quality and flavor profile of Yanbian yellow cattle through integrated metabolomic and biochemical approaches.

Therefore, this study aims to elucidate the major differences between the sexes of Yanbian yellow cattle and identify the key differential metabolites that affect meat quality and flavor. Understanding sex-specific differences in Yanbian yellow cattle is not only of scientific interest but also holds significant practical implications for the meat industry. The findings from this study could guide sex-specific processing protocols to maximize the suitability and quality of beef from different sexes for the development of various products and provide a scientific basis for market segmentation and consumer communication, enabling producers to better meet consumer preferences for tenderness, flavor, or nutritional profile.

## 2. Materials and Methods

### 2.1. Sample Preparation

Twelve Yanbian yellow cattle, six bulls and six cows, each group of similar weight, were purchased from Bao Ren Herding Co., Yanji, Jilin Province, China. All cattle were reared under the same pasture under the same feeding conditions until 32 ± 2 months of age. After slaughter, the *longissimus dorsi* (LD), *gluteus medius* (GM), and *triceps brachii* (TB) muscles were excised within 24 h, and each animal’s meat sample was randomly divided into six technical replicates, designated as bulls (BL) and cows (CL), transported to our laboratory at a controlled temperature of 4 °C, and promptly minced. After removal of fascia and connective tissue, the minced tissues were loaded into centrifuge tubes, rapidly frozen with liquid nitrogen, and stored at −80 °C for later analysis of volatile flavor compounds.

### 2.2. Moisture, Protein, Fat, Meat Color, and Shear Force

Moisture content was determined by drying samples to a constant weight at 105 °C (GB5009.3-2016) [18]. Protein content was assessed using the Kjeldahl method, applying a conversion factor of 6.25 (GB5009.5-2016) [19]. Fat content was measured through Soxhlet extraction using petroleum ether (GB5009.6-2016) [20]. Meat color was measured using a CR400 colorimeter (Konica Minolta Co., Tokyo, Japan) with CIE standard illuminant C as the light source, a 2° observer angle, and an 8 mm aperture diameter, calibrated with a standard white tile. Measurements were performed in triplicate on both surfaces of the muscle after a blooming time of 30 min, specifically on the external surface and the internal surface. Shear force was determined using a Texture Analyzer (Model TA. XTPlus, Stable Micro Systems, Godalming, UK) equipped with a Warner–Bratzler shear blade. Samples were sheared perpendicular to the muscle fiber orientation at a crosshead speed of 2.0 mm/s. Peak force (N) was recorded as the shear force value, with six replicates measured per treatment group. Prior to shear force testing, all samples were cooked in a water bath at 80 °C for 20 min.

### 2.3. Fatty Acids

Fatty acids were determined following a method devised by Dai et al. [21]. Briefly, muscle samples (2 g) were mixed with a chloroform–methanol solution (1:2, *v*/*v*). Triundecanoin was used as an internal standard solution, which could be stored and refrigerated for up to 1 month. Fatty acid methylation was carried out using the transesterification method. The upper organic phase was then collected and concentrated under a stream of nitrogen gas. Information regarding the standardized products is provided in Appendix A. The fatty acid composition was analyzed via GC, employing a Thermo Trace 1300 (Trace 1300, Thermo Fisher Scientific, Waltham, MA, USA) instrument fitted with a flame ionization detector. Helium was utilized as the carrier gas, maintaining a flow rate of 1 mL/min with a 100:1 split ratio. The temperature at the detector’s injection port was constantly set at 260 °C. The chromatographic conditions included an initial temperature of 170 °C, which was increased to 220 °C after 30 min, followed by an increment of 4 °C/min for 15 min, culminating in a temperature of 240 °C for an additional 20 min. The analysis was conducted using a sophisticated polyethylene glycol capillary column (SP-2560, dimensions: 100 m × 0.25 mm × 0.2 µm). Identification of fatty acids was achieved by comparing retention times with those of standard fatty acid samples (37 kinds of fatty acid methyl esters with accurately determined purity values were used as raw materials), with concentrations calculated based on peak areas. The findings were expressed as mg of fatty acid per 100 g of meat.

### 2.4. Determination of Amino Acids

The analysis of amino acid composition was conducted based on previously published methods with minor modifications [22]. Initially, 1 g sample was mixed with 0.2 mL of phenol and 10 mL of 6 M hydrochloric acid, subsequently hydrolyzed at 110 °C for a duration of 24 h. Tryptophan was not determined due to its degradation under acid hydrolysis conditions. The hydrolysates, once desiccated under a nitrogen atmosphere, were dissolved and reconstituted in a sodium citrate buffer (pH 2.2), then filtered through a 0.22 µm membrane. Analyses were conducted utilizing a Hitachi L8900 amino acid autoanalyzer (Tokyo, Japan). For amino acid standards, the mixed amino acid standard solution as specified in GB 5009.124-2016 [23] was employed. Before use, it was stored in a refrigerator at 4 °C, shielded from light. Details about the standardized products are presented in Appendix A. The composition of amino acids was identified by aligning the retention times with those of standards, and concentrations were quantified based on the corresponding peak areas, reported in mg per 100 g of meat.

### 2.5. Volatile Flavor Compounds (VOCs)

#### 2.5.1. E-Nose

Each meat sample (5 g) was enclosed within a headspace vial, subsequently equilibrated at a constant temperature of 40 °C for 30 min to prepare for analysis with an E-nose (PEN-3, Airsense Co., Ltd., Schwerin, Germany). Prior to analysis, the E-nose detector was calibrated. The sample vial was then correctly positioned on the testing tray, undergoing a cleaning cycle of 60 s followed by a measurement duration of 90 s at a consistent gas flow rate of 100 mL/min. Each sample was assessed in triplicate, with sensor readings captured at the 90 s mark. Analytical results were processed using Winmuster software (version 1.6.2, Airsense Analytics GmbH, Schwerin, Germany), with the sensor sensitivities delineated in Appendix A.

#### 2.5.2. GC-IMS

For the determination of the VOCs present in the samples, we employed a Flavour Spec GC-IMS system (1H1-00053, G.A.S., Dortmund, Germany). The analysis utilized a column of type WAX (15 m × 0.53 mm, 1 μm). Minced meat (2 g) was placed into a 20 mL headspace vial and subjected to 20 min of incubation at 60 °C, with an agitation speed of 500 r/min. The headspace injection needle temperature was maintained at 85 °C, with an injection volume of 1 mL administered in non-shunt mode. The IMS apparatus was regulated at a constant temperature of 45 °C. The drift gas flow was maintained at 150 mL/min, utilizing nitrogen with a purity of 99.999% as the carrier gas. The flow rate protocol was as follows: 0–2 min (2 mL/min), 2–10 min (10 mL/min), and 10–20 min (100 mL/min). Additionally, the drift gas (nitrogen, purity 99.999%) was kept at a flow rate of 150 mL/min. The qualitative assessment of distinctive VOCs was conducted using the NIST and IMS databases integrated within the GC-IMS Library Search software (version 0.4.10, G.A.S., Dortmund, Germany). Interpretive visualizations, including differential and fingerprint maps, were generated through the Reporter and Gallery Plot plugins (version 0.4.10) of the Laboratory Analytical Viewer analysis software (version 0.4.10).

#### 2.5.3. Relative Odor Activity Value

The evaluation of each VOCs’s contribution to flavor was conducted using the relative odor activity value (ROAV) method, as delineated in [24]. This method involved assigning a benchmark ROAV of 100 to the compound that showed the most significant influence on flavor. Subsequent ROAVs for other compounds were measured in relation to this benchmark. The calculation formula was as follows:
(1)$${\mathrm{ROAV}}_{n}\approx 100\times \frac{C_{n}\%}{C_{\mathrm{stan}}\%}\times \frac{T_{\mathrm{stan}}}{T_{n}}$$ where C_n_% and T_n_ denote the respective relative contents and perceptual thresholds of assorted volatile compounds. C_stan_% and T_stan_, conversely, refer to the relative content and threshold of the primary volatile compound, which showed the most significant influence on flavor perception.

### 2.6. Metabolite Extraction and LC-MS/MS Detection

For sample preparation, 100 mg of each sample was accurately weighed. Then, 200 μL of ice-cold water and 800 μL of a methanol–acetonitrile (1:1, *v*/*v*) solution were added. The mixture was thoroughly vortex-mixed and sonicated for 60 min with the tube immersed in an ice bath throughout the sonication process. Following this, 300 μL of a 20% methanol–acetonitrile solution, serving as an internal standard, was added, and the mixture was vortexed for 3 min, then centrifuged at 12,000 rpm for 10 min at 4 °C. Subsequently, 200 μL of the resultant supernatant was decanted into a correspondingly numbered centrifuge tube and maintained at −20 °C for 30 min. The sample was centrifuged again for 3 min using the same settings. Thereafter, 180 μL of the supernatant was transferred into a designated injection vial for subsequent analysis. For chromatographic separation, a Waters ACQUITY Premier HSS T3 Column (1.8 µm, 2.1 mm × 100 mm) was employed. The mobile phases were composed of 0.1% formic acid in water (phase A) and 0.1% formic acid in acetonitrile (phase B). The column temperature was initiated at 40 °C and the flow rate was set at 0.4 mL/min, with an injection volume of 4 µL. Data were logarithmically transformed to minimize the effects of both noise and the high variance of the variables. The format of the raw LC-MS/MS data was converted to “.mzXML” format by using ProteoWizard (version 4.4.2) (https://proteowizard.sourceforge.io/), and the LC-MS/MS data in “.mzXML” format were processed with a program developed in-house using R and XCMS for peak detection, extraction, alignment, and integration. Then, an MS/MS database developed in-house (BiotreeDB)was used for metabolite annotation.

### 2.7. Statistical Analysis

Data are expressed as the mean ± standard deviation (*n* = 6) derived from six biologically independent experimental replicates. Statistical analysis was performed using mixed-effects ANOVA with animal as a random effect to account for intra-animal correlation, and sex, muscle (anatomical location), and sex × muscle interaction as fixed effects. ANOVA, PCA, OPLS-DA, and KEGG metabolic pathway maps and regulatory network maps were analyzed and plotted using R software (version 4.4.2), and *p* < 0.05 indicated significance. The stability and reliability of the model were assessed through a permutation test of the OPLS-DA (PCA), with R^2^Y and Q^2^ intercept values approaching 1 serving as the evaluation criteria. Metabolites with significant differences were selected based on VIP > 1 and FDR < 0.05. Biological replicates and technical replicates were included in all analyses.

## 3. Results

### 3.1. Moisture, Protein, Fat, Meat Color, and Shear Force Analysis

As shown in Table 1, significant differences (*p* < 0.05) were observed in L* values among the three CL muscles: *triceps brachii* (33.53 ± 0.93), *longissimus*
*dorsi* (31.02 ± 0.98), and *gluteus medius* (32.06 ± 1.18). The L* values of the bull’s *gluteus medius* (35.87 ± 1.59) were significantly higher than those of the *triceps brachii* (30.44 ± 1.16) and *longissimus*
*dorsi* (29.86 ± 1.93) in BL.

For b* values in CL, the *longissimus dorsi* (10.39 ± 1.73) was significantly higher than the *triceps brachii* (9.30 ± 1.10) and *gluteus medius* (9.37 ± 1.62). In BL, the b* values of the *triceps brachii* (9.30 ± 1.10) and *gluteus medius* (9.37 ± 1.62) were slightly higher than that of the *longissimus dorsi* (8.85 ± 1.77), but not significantly.

Additionally, significant sex differences were found in the L* values of the *triceps brachii* and *gluteus medius* in Yanbian yellow cattle. Specifically, the L* value of the *triceps brachii* in CL was significantly higher than that in BL, while the L* value of the *gluteus medius* in CL was significantly lower than that in BL. No significant sex differences were observed in the other color indexes (a*, b*).

Shear force, a key indicator of meat tenderness, exhibits an inverse relationship with tenderness (lower values indicate greater tenderness). As presented in Table 1, significant differences (*p* < 0.05) were observed in shear force values among different muscle cuts of Yanbian yellow cattle. Both sexes showed the same tenderness ranking pattern across cuts: *triceps brachii* > *gluteus medius* > the *Longissimus dorsi*, indicating that the *longissimus dorsi* exhibited optimal tenderness. Notably, cows demonstrated significantly lower shear force values than bulls (*p* < 0.05), suggesting superior tenderness in female cattle.

### 3.2. Fatty Acids and Amino Acids

The fatty acid composition of Yanbian yellow cattle exhibited significant sex-related variations (Table 2). The analysis identified 20 fatty acids, comprising 10 saturated (SFAs) and 10 unsaturated (UFAs) fatty acids. *Longissimus* dorsi showed the greatest diversity (20 fatty acids), followed by *triceps brachii* (18). The BL samples contained 14 fatty acids collectively, while *gluteus medius* had the fewest (13). Significant SFA concentration differences occurred among CL cuts: *longissimus* dorsi (593.64 ± 52.80 mg/100 g), *gluteus medius* (405.22 ± 72.70 mg/100 g), and *triceps brachii* (203.19 ± 55.40 mg/100 g). Similarly, BL *longissimus* dorsi (225.15 ± 89.72 mg/100 g) and *triceps brachii* (225.10 ± 40.11 mg/100 g) exceeded *gluteus medius* (134.21 ± 45.13 mg/100 g). CL demonstrated a significantly higher total SFA content than BL.

Monounsaturated fatty acid (MUFA) concentrations also differed significantly across CL cuts: *longissimus* dorsi (656.33 ± 94.44 mg/100 g), *triceps brachii* (229.77 ± 35.56 mg/100 g), and *gluteus medius* (97.65 ± 17.6 mg/100 g). The total MUFA content was significantly higher in CL than BL, except in *gluteus medius*. Polyunsaturated fatty acid (PUFA) levels followed similar patterns: *longissimus* dorsi and *triceps brachii* in CL contained significantly more PUFAs than *gluteus medius*. BL showed no significant PUFA differences among cuts.

Beef provides crucial proteins and essential amino acids (EAAs). Beyond nutritional functions, amino acids generate flavor volatiles through Maillard reactions and Strecker degradation, significantly influencing meat texture and flavor [25]. Appendix A details sex-specific amino acid profiles in Yanbian yellow cattle, categorized as EAAs (Thr, Val, Ile, Leu, Phe, Lys, Met, His) and NEAAs (Asp, Ser, Glu, Gly, Ala, Cys, Tyr, Arg, Pro). Significant compositional variations occurred across cuts. The total amino acid (TAA) concentrations were highest in *gluteus medius* for both BL (19.80 ± 0.30 g/100 g) and CL (16.33 ± 0.03 g/100 g), exceeding *triceps brachii* (BL: 17.34 ± 0.86; CL: 15.25 ± 0.09 g/100 g) and *longissimus* dorsi (BL: 14.66 ± 0.99; CL: 13.47 ± 0.22 g/100 g) (*p* < 0.05). Similarly, EAA levels were highest in *gluteus medius* (BL: 7.47 ± 0.29 g/100 g; CL: 5.87 ± 0.13 g/100 g) versus *triceps brachii* (BL: 6.35 ± 0.25; CL: 5.59 ± 0.03 g/100 g) and *longissimus* dorsi (BL: 5.36 ± 0.40; CL: 5.04 ± 0.11 g/100 g) (*p* < 0.05). The EAA/TAA (35.95–37.70%) and EAA/NEAA (56.44–60.57%) ratios met FAO/WHO standards (40% and 60%, respectively), indicating balanced EAA composition.

### 3.3. E-Nose

Figure 1 shows the utility of the electronic nose in discerning volatile compound profiles in Yanbian yellow cattle across different sexes. The orthogonal partial least squares discriminant analysis (OPLS-DA) method exceled in modeling interactions among multiple dependent and independent variables, with a distinct advantage in its ability to exclude irrelevant variations in the independent variables [26].

This exclusion significantly consolidated categorical data into a principal component, thereby enhancing the model’s clarity and facilitating more pronounced visualization of discriminant effects and principal component score plots. Furthermore, the absence of overlap between the sample distributions of the BL and CL groups indicated substantial disparities in their volatile flavor compositions.

The electronic nose, equipped with odor detection sensors, was adept at identifying VOCs. Figure 2 shows the distinct response patterns of sensors to odors from Yanbian yellow cattle, differentiated by sex. The abundance of VOCs was markedly increased in CL compared to BL. Sensors R6, R7, and R8 showed distinct response degrees, showing the variability in VOCs, which predominantly encompassed aromatic hydrocarbons, alkanes, alcohols, aldehydes, ketones, and other aromatic compounds. However, the chemical structures of these substances have yet to be determined, necessitating further analysis through GC-IMS to elucidate the composition of these VOCs.

### 3.4. GC-IMS

Ion mobility spectroscopy was utilized to elucidate the variations in VOCs between different sexes of Yanbian yellow cattle. As depicted in Figure 3, the dot colors indicate concentration levels: red for higher concentrations and light blue for lower. This method evaluated the compounds through a comparative analysis of retention and drift times [27]. Notably, the reactive ion peak, delineated by a crimson vertical line at the 1.0 mark on the horizontal axis, corresponds to a normalized drift time of 7.83 s and a migration time ranging from 2.035 to 2.038 ms. When using the BL sample on the left as a reference, the significant escalation in signal intensity observed in the CL sample suggested pronounced disparities on VOCs between the two groups.

### 3.5. Metabolite Composition of Yanbian Yellow Cattle of Different Sexes

Figure 4A shows that the metabolites of male and female Yanbian cattle are discernible, though the 95% confidence intervals of these samples slightly overlap. This necessitated the employment of orthogonal partial least squares discriminant analysis (OPLS-DA) for further analysis [27]. The OPLS-DA outcomes distinctly segregated the metabolite profiles in the *Longissimus dorsi* muscle across sexes, indicating differential metabolites (Figure 4D). The stability and reliability of this model were affirmed by R^2^Y and Q^2^ intercept values nearing 1, as established through permutation tests of the OPLS-DA. Differential metabolites were screened based on the following criteria [28]: a VIP value exceeding 1, a *p*-value below 0.05, and LogFC surpassing 1 [29]. Adhering to these parameters, 201 metabolites were found to significantly vary between the CL and BL groups, with 109 being upregulated and 92 downregulated (Figure 4E). Subsequent hierarchical cluster analysis systematically classified these metabolites (Figure 4B,C).

## 4. Discussion

Differences in meat color between sexes were thus primarily evident through L* values, with no significant differences in a* and b* values. These results agree with findings by Mueller [30] and Daza [31], who investigated the effect of sex on meat quality in Avileña-Negra Ibérica and Angus × Nellore cattle, respectively. Mueller [30] suggested that differences in L* values are mainly related to intramuscular fat content. In the present study, the L* values of the *triceps brachii* and *longissimus* dorsi were higher in CL than in BL, while the L* value of the *gluteus medius* was lower in CL than in BL. This trend paralleled the observed differences in intramuscular fat content, indicating that higher fat content corresponds to higher L* values and brighter meat color.

The moisture content in both the *triceps brachii* and the *longissimus dorsi* showed no significant variance by sex. Only the moisture content of the *gluteus medius* in CL was significantly higher than that in BL, while there were no significant differences in other defined parts. This is consistent with the findings of a previous study, which indicated that there was no difference in moisture content between bulls and heifers of the same age [32]. No significant differences were observed in the protein levels across all examined muscles. In terms of lipid content, the fat content of the *longissimus dorsi* in CL was significantly higher than that in BL, while other comparisons between muscle types and sexes demonstrated no significant differences. Flavor substances in meat are mainly generated through fat oxidation, which significantly influences consumers’ sensory evaluation of meat and meat products [33]. It has also been demonstrated that there is a correlation between the fat content in meat and its juiciness, as the flavor substances released from fat stimulate saliva secretion in the mouth and enhance the juiciness of the food [34].

Our results demonstrating elevated lipid levels in the *triceps brachii* and *longissimus dorsi* muscles of female Yanbian yellow cattle align with those reported by Li et al. [32] in Simmental cattle. This consistency across breeds reinforces the notion that sexually dimorphic fat deposition, wherein females generally exhibit higher intramuscular fat content, represents a recurring phenomenon in bovines.

A study by Bureš and Bartoň [35] associated reduced tenderness in bulls with higher collagen and lower intramuscular fat content, a conclusion supported by our shear force measurements. We observed significantly higher shear force values (*p* < 0.05) in male than in female Yanbian yellow cattle, objectively confirming inferior tenderness in bull meat.

Sex serves as an important factor in muscle quality, with the hormonal milieu of cattle directly affecting the lipid and protein content within the musculature. Studies indicate that under identical age and nutritional conditions, elevated estrogen levels in CL (vs. BL) stimulate lipogenesis, while reduced androgen levels diminish fat synthesis inhibition, ultimately promoting greater fat deposition in CL [36].

To analyze the variances in VOCs, GC-IMS was used to generate fingerprint profiles of each sample, facilitating detailed comparisons, as illustrated in Figure 3. The consistency observed across these profiles indicated the method’s reproducibility. Although the VOC compositions of the samples were broadly analogous, marked disparities were evident in the concentrations of specific VOCs [37]. For instance, the BL sample predominantly featured isobutanol (distinctive odor), alongside 2-pentanone and tetrahydrofuran (sweet fruity aroma) and isoamyl alcohol (apple brandy). These compounds were significantly more concentrated than in the CL sample. In contrast, the CL sample exhibited higher concentrations of 3-hydroxy-2-butanone (creamy aroma), hydroxy acetone with its caramel scent, and 3-carene (pine aroma). Furthermore, there was a significant variation in the concentrations of certain VOCs compared to the BL group. VOCs like heptanal, methyl valerate, and octanal imparted fruity flavors, whereas hexanal, cis-4-heptenal, and 2-ethylpyrazine contributed greasy undertones. Additionally, pentanol and 2-heptanol introduced grassy notes, and 2-heptanone added a cheese-like aroma. In summary, the GC-IMS methodology proved highly effective in delineating significant differences in the levels of VOCs in Yanbian yellow cattle across different sexes. Table 3 revealed the identification of 35 different VOCs such as alcohols, ketones, aldehydes, and esters in the Yanbian yellow cattle sample.

Human olfactory sensitivity to different VOCs varies significantly, largely determined by the detection threshold, the minimum concentration at which an odorant is perceptible. To objectively evaluate aroma profiles, researchers utilize the odor activity value (OAV), which quantifies the ratio of a compound’s concentration in a sample to its odor threshold, thereby integrating both the compound’s presence and its perceptibility. Nonetheless, the absolute quantification of these compounds poses substantial challenges due to the complex nature of the samples, which typically contain diverse VOCs. Consequently, the relative odor activity value (ROAV) is commonly utilized to assess the contributions of individual volatile compounds to overall flavor perception. Analytically, compounds with an ROAV equal to or greater than 1 are considered primary contributors to a sample’s flavor profile. In comparative analyses of BL and CL sample groups, the ROAVs of 19 VOCs were measured, among which five—3-hydroxy-2-butanone-M, 3-hydroxy-2-butanone-D, isopentanol, acetone, and ethyl acetate—showed ROAVs exceeding 1 (Table 4). These findings align with recent research by Yao et al. [38], who identified similar flavor-active compounds in Tianzhu white yak. Our studies revealed significant variances in the concentrations of different VOCs between the BL and CL groups. These differences are hypothesized to stem from the influence of sex hormones on the VOCs. Our results are consistent with studies on Ningxiang and Berkshire pigs [27], using 2-pentylfuran, (E)-2-octenal, and pentanal, which were determined from ROAVs to be the “sweet–fruit–floral” characteristic markers of NX (and BN) pork. Acetic acid is a marker of sourness in BKS pork, and the ROAV results directly explain the sensory differences among different varieties of pork. Our results identified pentanal as the key aroma compound (ROAV > 1) in Yanbian yellow cattle, underscoring its universal role in defining the green and fatty aromatic notes in beef. Sex hormones modulate volatile organic compound (VOC) profiles primarily through lipid metabolism pathways, especially arachidonic acid and linoleic acid metabolism, which are critical precursors of key aroma compounds. Specifically, elevated estrogen levels in female cattle may promote lipid deposition and subsequent oxidation, leading to increased generation of aldehydes (e.g., hexanal, valeraldehyde) and alcohols (e.g., 1-octen-3-ol, hexanol)—compounds widely associated with desirable fatty and green aroma notes in beef [39]. Conversely, androgens in bulls may favor lean tissue development, thereby limiting substrate availability for these flavor-forming pathways.

Of these VOCs, compounds such as hexanal (green, grassy) and 1-octen-3-ol (mushroom-like) are particularly influential in shaping consumer perception due to their low odor thresholds and frequent association with positive sensory attributes in cooked beef [40]. The higher abundance of these compounds in female Yanbian yellow cattle likely contributes to their distinct and preferred flavor profile.

Sex factors specifically regulate growth hormone (GH) secretion, which is important in regulating animal growth and production and induces lipolysis [41], while estradiol, a form of estrogen, increases GH release and decreases the release of the growth inhibitory factor (SRIF) from the anterior pituitary cells of bovine cows [42]. Furthermore, Ardiyanti et al. [43] reported that the GH allele C was linked to increased unsaturated fatty acids in heifers of Japanese Black cattle. Our results also showed a higher proportion of unsaturated fatty acids in females, particularly in the longissimus dorsi muscle. However, some divergence was noted in saturated fatty acid profiles, which may reflect breed-specific genetic or nutritional factors.

From the perspective of lipid anabolism, glycerides—which serve as energy storage molecules—constitute one of the biosynthetic pathways for glycerophospholipid production. Estrogens are known to enhance the activity of phosphatidylinositol 3-kinase (PI3K), thereby promoting the intracellular synthesis of phosphatidylinositol, which contributes to membrane integrity and facilitates metabolic signal transduction. Androgens, on the other hand, may modulate triglyceride synthesis by regulating glycerol-3-phosphate dehydrogenase (GPDH) activity, ultimately influencing glycerophospholipid levels and metabolism in adipocytes [44]. The biosynthesis of triglycerides (TGs) involves several intermediate lipids, including lysophosphatidic acid, phosphatidic acid (PA), and diacylglycerol (DAG). In the present study, the contents of phosphatidic acid, phosphatidylcholine, and phosphatidylethanolamine in the *longissimus*
*dorsi* muscle of cows were significantly upregulated compared with bulls. Concurrently, estrogen levels were significantly higher, while androgen levels were significantly lower in cows. These findings provide mechanistic evidence that sex hormones modulate lipid anabolic pathways by regulating the glycerophospholipid metabolic network, which is consistent with the results reported by Oh et al. [44]. Their study demonstrated that testosterone exerts an anti-adipogenic effect through the inhibition of glycerol-3-phosphate dehydrogenase (GPDH) activity, whereas 17β-estradiol and progesterone promote adipogenesis.

The concentration of alcohols and aldehydes in the CL samples was significantly higher than in the BL samples, whereas the ketone levels were notably lower in the CL samples, with acetone being predominant in both groups. Aldehydes, derived primarily from the oxidative breakdown of fatty acids, play a crucial role in meat flavor, particularly in ruminant meats [45]. They engage in reactions with amino acids and contribute significantly to the aroma due to their potent scent and low sensory threshold [46]. However, at higher concentrations, aldehydes can impart off-flavors, such as rancidity. Certain aldehydes, like hexanal, heptanal, octanal, nonanal, and 3-methylbutanal, are particularly significant for their distinct aromas, ranging from fatty and grassy to fruity [36]. Alcohols, chiefly produced through the degradation of conjugated linoleic acid in muscle, also form a significant portion of beef’s volatile profile. Pentanol and n-butanol were the primary alcohols identified. Although they possess a higher odor threshold, their impact on flavor was generally less pronounced than that of aldehydes. This study also highlighted a correlation between the higher aldehyde and alcohol content in the CL group with the variance in fatty acid composition, suggesting that fatty acid composition might drive these differences in flavor profiles. Ketones are typically produced from lipid degradation, alcohol oxidation, and the catabolism of esters. Despite having higher thresholds than aldehydes, ketones contribute stable and pleasant aromas, typically described as fruity, buttery, and creamy. The BL group exhibited a higher concentration of certain ketones, such as 2-butanone and acetone, known for their pleasant fruity aroma yet solvent-like nature. Conversely, 3-hydroxy-2-butanone was more prominent in the CL group, contributing a creamy note to the flavor profile. Unlike aldehydes and alcohols, ketone levels did not correlate with fatty acid differences, suggesting that alternative mechanisms influence their distribution.

In the analysis comparing the differential metabolites in the *dorsal longissimus* muscle of Yanbian yellow cattle across sexes, this study primarily examined amino acids and their derivatives, such as L-Tyrosyl-L-Asparagine, Lys-Tyr-Val-Lys, and L-Tryptophanamide. Additionally, this study encompassed hormones and hormone-related compounds, such as testosterone and estradiol. This analysis revealed that sex hormones significantly influence the variance in metabolic substances between the sexes in Yanbian yellow cattle. Hormones are crucial in regulating animal growth, development, and nutrient metabolism, particularly sex hormones, which exhibit substantial variations between bulls and heifers. A previous study [47] confirmed that testosterone levels were markedly reduced in the metabolites from the heifers’ *Longissimus dorsi* muscles, whereas β-estradiol-3-sulfate levels were notably increased. As a downstream product of β-estradiol [48], β-estradiol-3-sulfate, alongside testosterone, constitutes the primary active elements of androgens and estrogens, respectively. Sex hormones impact protein synthesis in skeletal muscle, tissue development, including muscle and fat, and overall carcass quality [49]. Specifically, androgens enhance protein synthesis and skeletal muscle growth, augment energy expenditure, and promote lipolysis and testosterone inhibits lipoprotein esterase activity, decreasing triglyceride synthesis and increasing the breakdown of triglycerides into free fatty acids [50]. Furthermore, this study identified significant variations in prostaglandin (PG) levels among the metabolites, with PG directly stimulating hormone-releasing neuroendocrine cells, thus enhancing the secretion of hormones such as luteinizing hormone, which also boosts the production of follicle-stimulating hormone and, consequently, testosterone.

The metabolic pathways exhibiting notable distinctions encompass glycerophospholipid metabolism, retrograde endocannabinoid signaling, α-linolenic acid metabolism, arachidonic acid metabolism, ether lipid metabolism, autophagy in animals, glycosylphosphatidylinositol anchor biosynthesis, phosphonate and phosphinate metabolism, the sphingomyelin signaling pathway, and the metabolism of alanine, aspartate, glutamate, sphingolipids, and histidine, among others (Figure 5).

The clear sex-dependent differences in meat quality traits observed in this study provide a scientific basis for strategic breeding and management. Breeders could prioritize the rearing of female Yanbian yellow cattle for the production of premium beef with superior tenderness, targeting high-end markets. Conversely, bulls could be selected for efficient lean meat production, where yield is the primary objective. Genetic selection programs could further incorporate markers associated with favorable fat deposition and flavor profiles.

This enables the development of targeted marketing strategies, such as labeling beef from females as “premium tender” or “enhanced flavor” products, thereby adding value and meeting diverse consumer preferences. The industry can optimize processing protocols based on sex to maximize the quality of the final product.

## 5. Conclusions

This study analyzed the differences in metabolites and VOCs between sexes for Yanbian yellow cattle using GC-IMS and LC-MS/MS. Additionally, we determined their moisture content, protein content, fat content, meat color, and shear force. Sex differences influenced endogenous hormones, including sex hormones, which in turn altered metabolic pathways, such as lipid metabolism, fatty acid biosynthesis, amino acid metabolism, and glycolysis. This study represents the first comprehensive investigation into the effects of sex on meat quality and flavor profiles in Yanbian yellow cattle while also elucidating the regulatory roles of hormones and associated metabolic pathways. These findings provide a scientific basis for gender-based slaughter strategies and support the targeted processing of beef from different carcass segments.

## Figures and Tables

**Figure 1 foods-14-03175-f001:**
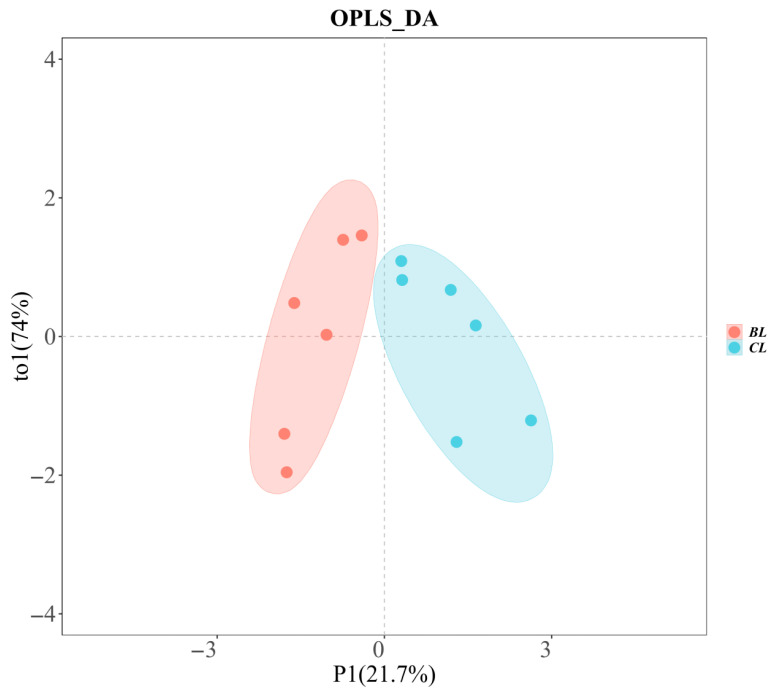
OPLS-DA loading plot of the electronic nose for Yanbian yellow cattle beef of different sexes.

**Figure 2 foods-14-03175-f002:**
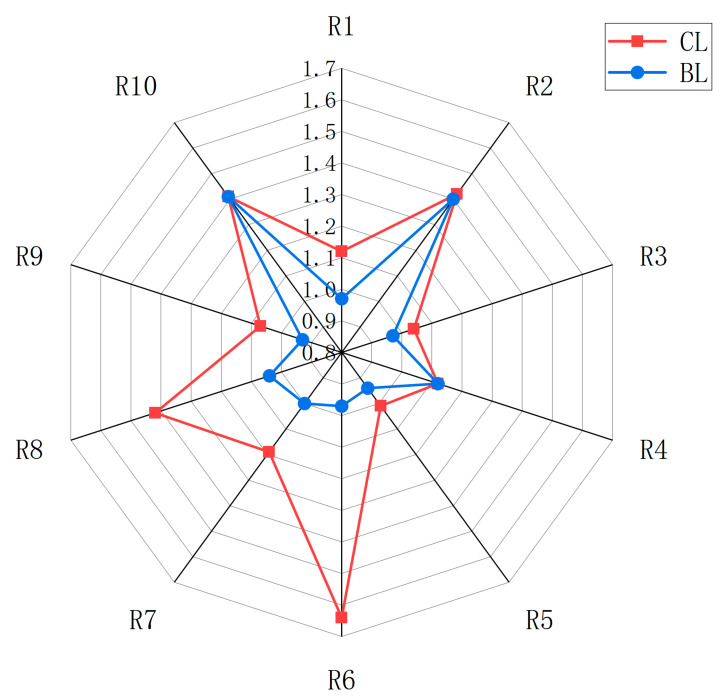
Radar chart of the electronic nose for Yanbian yellow cattle meat of different sexes.

**Figure 3 foods-14-03175-f003:**
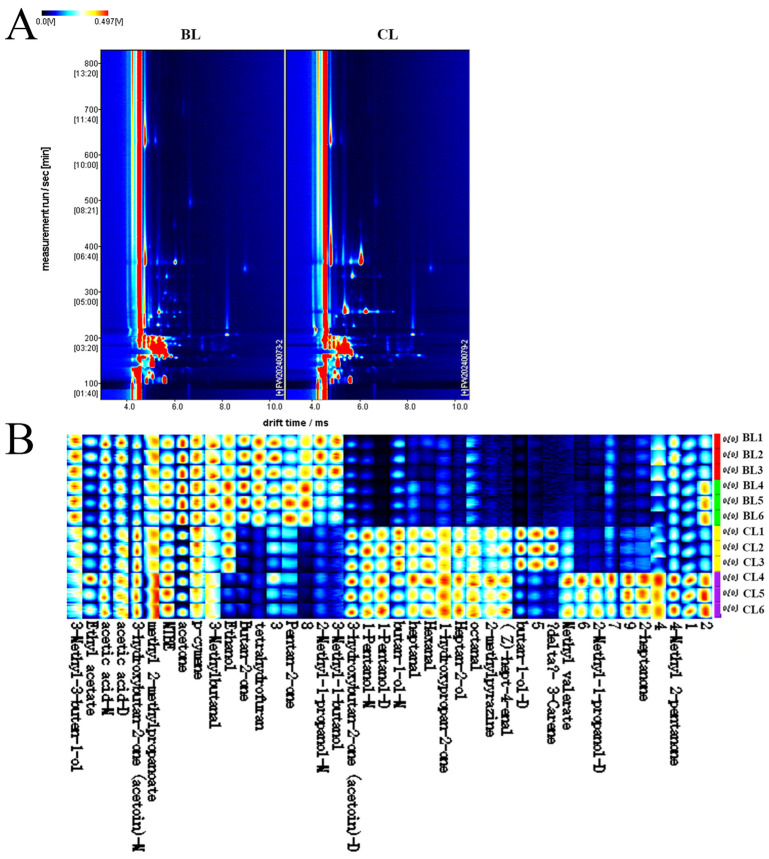
(**A**) Ion migration spectra for Yanbian yellow cattle of different sexes; (**B**) fingerprint of total volatile compounds for Yanbian yellow cattle beef of different sexes.

**Figure 4 foods-14-03175-f004:**
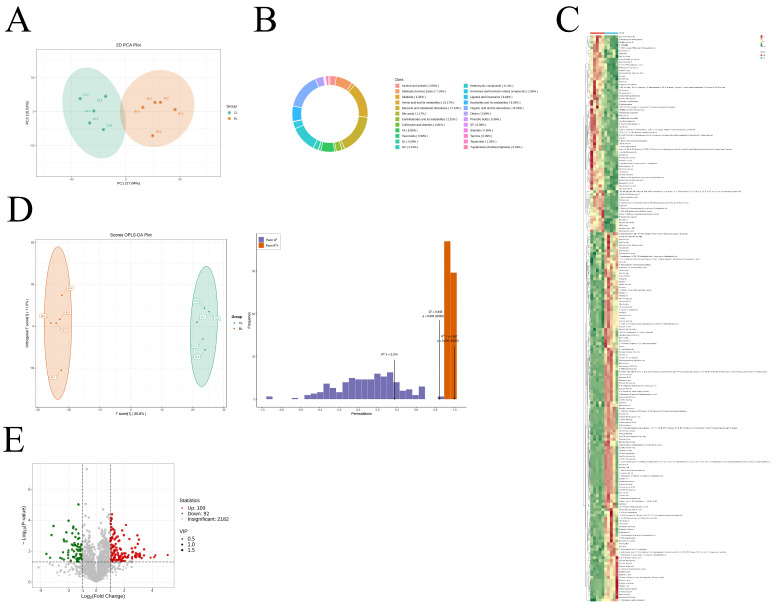
(**A**) PCA loading plot of the *longissimus dorsi* muscle of Yanbian yellow cattle of different sexes; (**B**) classification diagram of differential metabolites in the *longissimus dorsi* muscle of Yanbian yellow cattle of different sexes; (**C**) heat map of metabolite differences in Yanbian yellow beef between sexes; note: Vertical is clustering of metabolites, and different colors represent different values obtained after normalization transformation of the relative content of differential metabolites (red for high content, green for low content).(**D**) OPLS-DA loading plot and verification diagram of the *longissimus dorsi* muscle of Yanbian yellow cattle of different sexes; (**E**) volcano plot of differential metabolites in the *longissimus dorsi* muscle of Yanbian yellow cattle of different sexes.

**Figure 5 foods-14-03175-f005:**
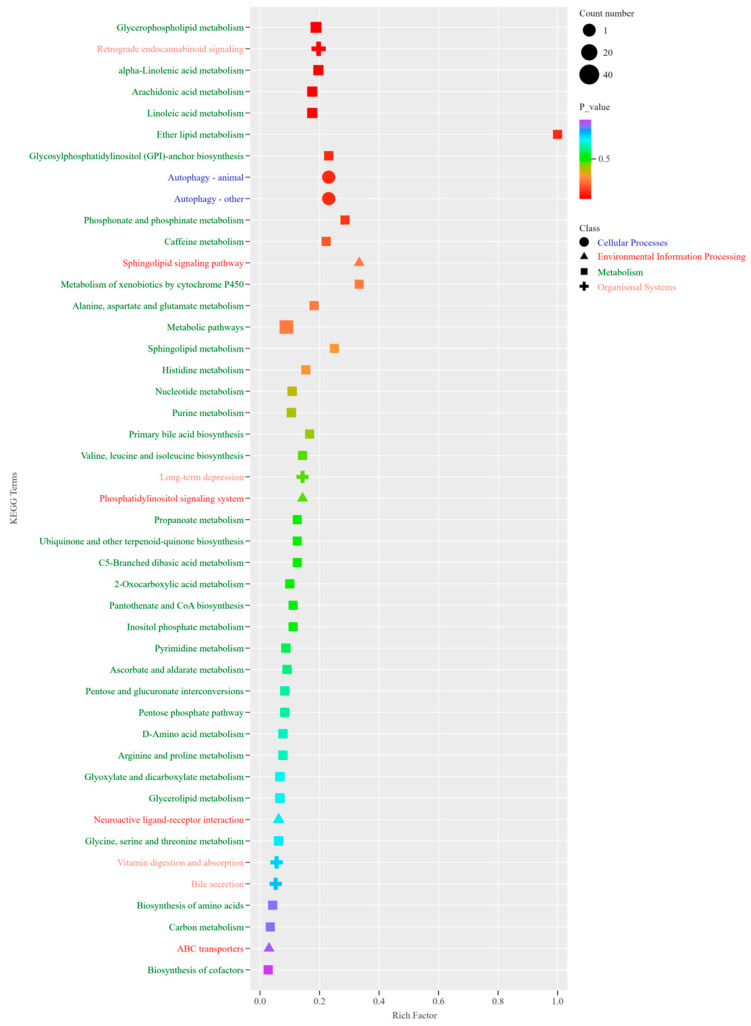
KEGG metabolic pathway enrichment diagram of significant differential metabolites.

**Table 1 foods-14-03175-t001:** Effect of gender on the quality of different parts of Yanbian yellow cattle.

Attribute	Cow			Bull		
	*Triceps brachii*	*Longissimus dorsi*	*Gluteus medius*	*Triceps brachii*	*Longissimus dorsi*	*Gluteus medius*
Moisture (%)	74.93 ± 1.23 ^a^	70.80 ± 1.78 ^c^	73.48 ± 1.26 ^b^	75.06 ± 1.24 ^a^	70.34 ± 1.96 ^c^	71.98 ± 0.93 ^b^*
Protein (%)	20.31 ± 1.52 ^a^	21.43 ± 1.74 ^a^	22.46 ± 1.38 ^a^	21.55 ± 0.81 ^a^	22.65 ± 1.14 ^a^	22.48 ± 1.24 ^a^
Fat (%)	5.25 ± 1.66 ^b^	9.69 ± 2.29 ^a^	3.53 ± 1.05 ^c^	4.02 ± 0.89 ^a^	4.74 ± 0.80 ^a^*	4.31 ± 0.91 ^a^
*L**	33.53 ± 0.93 ^a^	31.02 ± 0.98 ^b^	32.06 ± 1.18 ^ab^*	30.44 ± 1.16 ^b^*	29.86 ± 1.93 ^b^	35.87 ± 1.59 ^a^
*a**	12.87 ± 1.71 ^a^	12.36 ± 1.58 ^a^	13.16 ± 1.32 ^a^	11.48 ± 1.93 ^a^	11.19 ± 1.87 ^a^	13.73 ± 1.99 ^a^
*b**	9.30 ± 1.10 ^b^	10.39 ± 1.73 ^a^	9.51 ± 1.73 ^b^	9.50 ± 1.80 ^a^	8.85 ± 1.77 ^a^	9.37 ± 1.62 ^a^
Shear force(N)	182.03 ± 14.34 ^a^*	115.54 ± 8.92 ^c^*	146.16 ± 9.16 ^b^*	235.71 ± 15.48 ^a^	146.51 ± 14.16 ^c^	188.62 ± 8.61 ^b^

Note: ^a–c^ represent significant differences between sites of the same sex (*p* < 0.05); * represents significant differences between sexes of the same site (*p* < 0.05).

**Table 2 foods-14-03175-t002:** Fatty acid content of Yanbian yellow cattle of different sexes (mg/100 g).

Number	Fatty Acid Type	*Triceps Brachii*		*Longissimus Dorsi*		*Gluteus Medius*	
		Cow	Bull	Cow	Bull	Cow	Bull
1	C6:0	3.44 ± 0.43 ^b^	3.15 ± 0.45 ^a^	4.06 ± 0.31 ^a^	3.58 ± 0.93 ^a^	3.88 ± 0.3 ^ab^	3.326 ± 0.70 ^a^
2	C10:0	ND	ND	1.06 ± 0.16	ND	ND	ND
3	C11:0	1.201 ± 0.23 ^b^*	1.52 ± 0.12 ^a^	1.65 ± 0.31 ^a^	1.56 ± 0.19 ^a^	1.30 ± 0.23 ^b^	1.43 ± 0.24 ^a^
4	C12:0	0.58 ± 0.55	ND	1.31 ± 0.15	ND	ND	ND
5	C14:0	29.69 ± 6.54 ^b^	11.41 ± 2.99 ^a^*	43.65 ± 5.62 ^a^	11.46 ± 4.01 ^a^*	11.81 ± 3.96 ^c^	1.59 ± 0.87 ^b^*
6	C14:1	3.69 ± 1.15 ^b^	2.19 ± 0.77 ^a^	7.63 ± 2.12 ^a^	2.09 ± 0.76 ^a^*	1.32 ± 0.28 ^c^	0.81 ± 0.55 ^b^
7	C15:0	2.45 ± 0.75 ^b^	2.16 ± 0.74 ^a^	5.26 ± 0.99 ^a^	1.94 ± 0.95 ^a^*	1.15 ± 0.26 ^b^	0.97 ± 0.63 ^a^
8	C16:0	232.25 ± 43.94 ^ab^	126.87 ± 19.51 ^a^*	269.72 ± 26.31 ^a^	111.01 ± 47.77 ^a^*	18.31.13 ± 11.74 ^b^*	70.81 ± 47.08 ^a^
9	C16:1	17.53 ± 8.43 ^c^	12.3 ± 6.50 ^a^	56.85 ± 6.16 ^a^	14.38 ± 6.51 ^a^*	28.80 ± 3.67 ^b^	10.18 ± 5.57 ^a^*
10	C17:0	5.32 ± 2.91 ^b^	4.26 ± 1.64 ^a^	13.52 ± 1.40 ^a^	4.74 ± 1.80 ^a^*	1.55 ± 0.23 ^c^*	3.54 ± 1.14 ^a^
11	C17:1	4.28 ± 2.39 ^b^	3.00 ± 1.21 ^a^	7.46 ± 0.70 ^a^	3.06 ± 1.13 ^a^*	ND	2.59 ± 1.46 ^a^
12	C18:0	130.23 ± 19.9 ^b^	86.15 ± 10.0 ^a^*	241.87 ± 22.49 ^a^	70.87 ± 34.65 ^b^*	3.60 ± 0.80 ^c^*	6.77 ± 0.67 ^c^
13	C18:1n9t	3.51 ± 0.38 ^b^	3.72 ± 1.03 ^a^	19.28 ± 1.43 ^a^	5.29 ± 3.14 ^a^*	4.01 ± 1.08 ^b^	3.20 ± 1.86 ^a^
14	C18:1n9c	133.67 ± 24.27 ^b^	139.20 ± 36.11 ^a^	561.91 ± 85.5 ^a^	170.56 ± 72.9 ^a^*	77.28 ± 22.8 ^c^*	112.19 ± 36.61 ^a^
15	C18:2n6c	29.24 ± 7.42 ^b^	18.77 ± 3.94 ^a^*	36.03 ± 3.66 ^a^	19.23 ± 4.10 ^a^*	ND	12.21 ± 8.35 ^b^
16	C20:0	ND	ND	1.75 ± 0.12	ND	ND	ND
17	C20:1	1.04 ± 0.16 ^b^	ND	1.61 ± 0.28 ^a^	ND	ND	ND
18	C18:3n	1.21 ± 0.23 ^b^	ND	1.68 ± 0.19 ^a^	ND	ND	ND
19	C20:3n6	2.97 ± 0.61 ^b^	ND	3.30 ± 0.31 ^ab^	ND	3.57 ± 0.99 ^a^	ND
20	C20:4n6	12.32 ± 2.62 ^a^	5.88 ± 2.22 ^a^*	7.40 ± 0.83 ^b^	5.32 ± 0.58 ^a^*	14.28 ± 3.17 ^a^	4.53 ± 2.09 ^a^*
	SFA	405.22 ± 72.70 ^b^	225.10 ± 40.11 ^ab^*	593.64 ± 52.80 ^a^	225.15 ± 89.72 ^a^*	203.19 ± 55.40 ^c^	134.21. ± 45.13 ^b^
	MUFA	229.77 ± 35.56 ^b^	154.33 ± 30.31 ^b^*	656.33 ± 94.44 ^a^	217.24 ± 84.40 ^a^*	97.65 ± 17.60 ^c^	115.14 ± 51.79 ^b^
	PUFA	41.74 ± 5.76 ^a^	25.58 ± 6.17 ^a^*	49.65 ± 4.64 ^a^	26.08 ± 6.49 ^a^*	20.63 ± 3.04 ^b^	18.76 ± 7.61 ^a^

Note: ^a–c^ represent significant differences between cattle from different sites but of the same sex (*p* < 0.05); * represents significant differences between cattle of different sexes from the same site (*p* < 0.05). ND means not detected.

**Table 3 foods-14-03175-t003:** Relative content of volatile substances of Yanbian yellow cattle with different sexes.

Substance Name	CAS	Retention Index (RI)	Retention Time (Rt)/s	Drift Time (Dt)/min	Relative Peak Area (%)		*p* Value
					BL	CL	
1-Pentanol-D	C71410	1254.5	336.073	1.51165	0.08 ± 0.01	1.67 ± 0.13	<0.05
1-Pentanol-M	C71410	1255.6	336.979	1.25331	1.03 ± 0.08	4.42 ± 0.10	<0.05
2-Heptanol	C543497	1335.7	415.728	1.37889	0.11 ± 0.00	0.44 ± 0.06	<0.05
2-Methyl-1-propanol-D	C78831	1096.2	228.482	1.17253	0.34 ± 0.07	0.15 ± 0.01	<0.05
2-Methyl-1-propanol-M	C78831	1092.7	226.943	1.35747	0.11 ± 0.03	0.38 ± 0.24	<0.05
n-Butanol-D	C71363	1144.7	257.188	1.37751	0.34 ± 0.08	3.87 ± 1.74	<0.05
n-Butanol-M	C71363	1147.7	259.117	1.18109	1.67 ± 0.22	3.93 ± 0.73	<0.05
ethanol	C64175	981.2	184.157	1.04303	1.10 ± 0.04	0.81 ± 0.37	>0.05
3-Methyl-3-buten-1-ol	C763326	1251.1	333.358	1.16878	0.20 ± 0.01	0.17 ± 0.02	>0.05
CIS-4-heptenol	C6728310	1253.4	335.197	1.61856	0.07 ± 0.00	0.42 ± 0.03	<0.05
isopentanol	C123513	1206.3	298.938	1.23976	0.24 ± 0.03	0.10 ± 0.01	<0.05
**sterols**					5.35 ± 0.46	16.4 ± 2.59	<0.05
2-Heptanone	C78933	898.2	162.5	1.24903	0.17 ± 0.03	0.45 ± 0.23	<0.05
2-Butanone	C110430	1144.7	257.169	1.25255	10.54 ± 0.50	2.73 ± 0.38	<0.05
2-Pentanone	C107879	981.1	184.128	1.37244	0.18 ± 0.01	0.12 ± 0.00	<0.05
3-Hydroxy-2-butanone-D	C513860	1289.7	366.128	1.3332	3.54 ± 0.22	11.2 ± 0.92	<0.05
3-Hydroxy-2-butanone-M	C513860	1293.5	369.482	1.05432	11.84 ± 0.22	18.15 ± 0.51	<0.05
acetone	C67641	812.7	142.855	1.12104	43.70 ± 0.75	25.04 ± 0.37	<0.05
hydroxyacetone	C116096	1290.6	366.912	1.23909	0.14 ± 0.00	0.36 ± 0.00	<0.05
4-Methyl-2-pentanone	C108101	1014.1	194.996	1.18292	4.68 ± 0.46	4.97 ± 0.55	>0.05
**ketone**					74.82 ± 0.36	63.07 ± 2.08	<0.05
3-Methylbutyraldehyde	C590863	921.5	168.299	1.18256	6.00 ± 0.11	5.24 ± 0.33	<0.05
heptanal	C111717	1184.1	283.23	1.32767	0.21 ± 0.04	0.58 ± 0.04	<0.05
N-hexanal	C66251	1091.1	226.239	1.26743	0.21 ± 0.06	1.08 ± 0.09	<0.05
n-octanal	C124130	1292.3	368.474	1.40462	0.38 ± 0.03	0.80 ± 0.07	<0.05
**aldehyde**					6.82 ± 0.14	7.69 ± 035	<0.05
Methyl valerate	C624248	1093.5	227.298	1.56413	0.05 ± 0.01	0.25 ± 0.07	<0.05
Propyl 2-methylpropionate	C547637	929.6	170.383	1.14595	4.24 ± 0.45	4.45 ± 1.26	>0.05
ethyl acetate	C141786	879.3	157.943	1.34077	0.48 ± 0.10	0.72 ± 0.15	<0.05
**esters**					4.79 ± 0.57	5.44 ± 1.42	>0.05
Acetic acid-D	C64197	1485.6	633.257	1.14937	0.43 ± 0.03	0.47 ± 0.04	>0.05
Acetic acid-M	C64197	1486.3	634.511	1.05296	8.34 ± 0.18	8.76 ± 0.60	>0.05
**acidic**					8.78 ± 0.21	9.23 ± 0.55	>0.05
p-umbelliferyl hydrocarbons	C99876	1290.9	367.146	1.16878	0.81 ± 0.01	0.71 ± 0.04	<0.05
Methyl tert-butyl ether	C1634044	673.3	115.787	1.12559	2.52 ± 0.24	2.81 ± 0.55	>0.05
3-Carene	C13466789	1145	257.36	1.62654	0.06 ± 0.00	0.25 ± 0.16	<0.05
tetrahydrofuran (THF)	C109999	871.6	156.111	1.22905	0.70 ± 0.04	0.18 ± 0.01	<0.05
2-Methylpyrazine	C109080	1253.7	335.435	1.39246	0.09 ± 0.00	0.26 ± 0.03	<0.05
**other**					4.20 ± 0.23	4.23 ± 0.37	>0.05

Note: D represents dimers and M represents monomers.

**Table 4 foods-14-03175-t004:** ROAVs and the contribution of key odorant compounds to Yanbian yellow cattle.

Odorant	Threshold Value (μg/kg)	ROAV		Odor Description
		BL	CL	
3-Hydroxy-2-butanone-M	140	100	100	Buttery, Green
3-Hydroxy-2-butanone-D	140	29.8986	61.7080	Buttery, Green
1-Pentanol-D	150.2	0.0630	0.8576	Green
1-Pentanol-M	150.2	0.8109	2.2699	Green
n-Butanol-D	480	0.0838	0.7103	Alcohol
n-Butanol-M	480	0.4114	0.6315	Alcohol
3-Methyl-3-buten-1-ol	363,000	0.0001	0.0001	Sweet, fruity
isopentanol	4	7.0946	1.9284	Apple, Brandy
2-Heptanone	83,000	0.0002	0.0004	
2-Pentanone	1380	0.0150	0.0150	Fruity
acetone	832	6.2106	2.3271	Nutty, Bitter
Hydroxyacetone	10,000	0.0017	0.0028	Nutty, Bitter
4-Methyl-2-pentanone	138,000	0.0040	0.0028	Sweet, fruity
Ethyl acetate	5	11.3514	11.1074	Fruity, Sweet
Acetic acid-D	99,000	0.0005	0.0004	Sour
Acetic acid-M	99,000	0.0005	0.0004	Sour
Ethanol	950,000	0.0001	0.0001	Alcohol
Methyl valerate	110	0.0537	0.1753	Fruity
Propyl 2-methylpropionate	183,000	0.0027	0.0019	Fruity

## Data Availability

The original contributions presented in this study are included in the article/Appendix A. Further inquiries can be directed to the corresponding authors.

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
