# Peer review of "A Comprehensive Study of Meat Quality and Flavor Characteristics of Different Sexes of Yanbian Yellow Cattle Using GC-IMS and LC-MS/MS Technologies"

_foods, 2025, doi:10.3390/foods14183175_

Round 1
Reviewer 1 Report
Comments and Suggestions for Authors
The manuscript is well-structured and investigates an interesting and relevant topic in food science. The combined use of GC-IMS, LC-MS/MS, and E-nose provides valuable insights into sex-related differences in beef quality and flavor. However, there are some areas that require improvement in terms of clarity, scientific rigor, and interpretation of results.
- The manuscript states that this is the first report on sex differences in Yanbian yellow cattle. While this is valuable, the novelty should be emphasized more clearly in the introduction by comparing with previous studies on other cattle breeds. Consider explaining how these findings can be applied in meat industry practices (e.g., breeding strategies, processing suitability, or consumer preference).
-
Materials and Methods: The sample size is relatively small (6 bulls and 6 cows). Please justify whether this number is statistically sufficient for metabolomics and VOC profiling.
- More details should be provided on statistical methods: Were multiple testing corrections applied when identifying significant metabolites? How were PCA/OPLS-DA validated to avoid overfitting? Clarify whether biological replicates and technical replicates were included in all analyses (especially GC-IMS and LC-MS/MS).
- The discussion on ROAV values is important, but it would benefit from comparing with existing studies on beef or other livestock species. Consider reorganizing the results to reduce repetition (e.g., meat color, moisture, and fat content are described twice with similar explanations).
Results and Discussion: Some sections are descriptive but lack deeper biological or mechanistic interpretation. For example, the link between sex hormones, lipid metabolism, and flavor compounds should be explained with stronger evidence from literature.
- The discussion on ROAV values is important, but it would benefit from comparing with existing studies on beef or other livestock species. Consider reorganizing the results to reduce repetition (e.g., meat color, moisture, and fat content are described twice with similar explanations).
-
The conclusion section should be more concise and focus on the main findings. Currently, it restates much of the results. Instead, highlight the practical implications (e.g., how sex-based differences might guide meat processing or product development).
Some improvements are needed.
Author Response
Dear Editor and Reviewers:
We would like to thank the reviewers for carefully reading our manuscript.We appreciate the comments and suggestions.In the following, we include a point-by-point response to the comments from each reviewer. In the revised manuscript, all the changes have been highlighted in red.

Reviewer 2 Report
Comments and Suggestions for Authors
Dear Authors,
My comments are attached.

The English language and grammar should be improved.
Author Response

(The authors gave the same response as above.)

Round 2
Reviewer 2 Report
Comments and Suggestions for Authors
Dear Authors,
Thank you for the revised manuscript that shows significant improvements. I have a few minor comments:
- In Tables 1 and 2, please delete "The" from "The Longissimus dorsi"
- Line 329: It should be 3.3. ("3.4. GC-IMS") and "3.5. Metabolite composition of Yanbian yellow cattle of different sexes" should be 3.4.
- There are some formatting issues with the table names - please check.
Author Response
Dear Editor and Reviewers,
We are grateful for the careful review of our manuscript and appreciate the insightful comments and suggestions provided. Below, we have prepared a point-by-point response to the reviewers' comments. All corresponding revisions in the manuscript have been highlighted in red for ease of review.
Comment 1: In Tables 1 and 2, please delete "The" from "The Longissimus dorsi"
Response 1: We thank the reviewer for this helpful correction. We have carefully reviewed the manuscript and removed the article “the” preceding “Longissimus dorsi” throughout all tables, including Supplementary Table S2.
Comment 2: Line 329: It should be 3.3. ("3.4. GC-IMS") and "3.5. Metabolite composition of Yanbian yellow cattle of different sexes" should be 3.4.
Response 2: We thank the reviewer for attentively identifying these numbering errors. The section headings have now been corrected to ensure proper sequential order: “3.3. GC–IMS” and “3.4. Metabolite composition of Yanbian yellow cattle of different sexes”.
Comment 3: There are some formatting issues with the table names - please check.
Response 3: We thank the reviewer for highlighting these formatting issues. We have carefully reviewed the formatting of all tables and figures, including those in the supplementary materials, and have adjusted them to conform to the journal's author guidelines.